# NeurInt-Learning Interpolation by Neural ODEs

**Avinandan Bose**[*][†]
Indian Institute of Technology Kanpur
avibose@iitk.ac.in

**Aniket Das**[*]
Indian Institute of Technology Kanpur
aniketd@iitk.ac.in

**Yatin Dandi**
Indian Institute of Technology Kanpur
yatind@iitk.ac.in

**Piyush Rai**
Indian Institute of Technology Kanpur
piyush@cse.iitk.ac.in

## Abstract

A range of applications require learning image generation models whose latent space effectively captures the high-level factors of variation in the data distribution, which can be judged by its ability to interpolate between images smoothly. However, most generative models mapping a fixed prior to the generated images lead to interpolation trajectories lacking smoothness and images of reduced quality. We propose a novel generative model that learns a flexible non-parametric prior over interpolation trajectories, conditioned on a pair of source and target images. Instead of relying on deterministic interpolation methods like linear or spherical interpolation in latent space, we devise a framework that learns a distribution of trajectories between two given images using Latent Second-Order Neural Ordinary Differential Equations. Through a hybrid combination of reconstruction and adversarial losses, the generator is trained to map the sampled points from these trajectories to sequences of realistic images of improved quality that smoothly transition from the source to the target image.

## 1  Introduction

In the past few years, deep generative models' incredible success has demonstrated their ability to represent the underlying factors of variations in high dimensional data, such as images via low dimensional latent variables. These factors of variation are commonly visualized by interpolating between images by traversing particular paths in the latent space. Given any two images, it is often desirable to obtain a distribution over various possible trajectories of smooth and realistic image space interpolations. Learning such distribution would allow a more extensive analysis of the factors of variation in data. In this work, we propose an approach that jointly trains an encoder and a generator to successively transform latent vectors on a trajectory to interpolations from a given source to a target image. To flexibly model distributions over trajectories of latent vectors, we parameterize their dynamics in continuous-time using Neural Ordinary Differential Equations [5]. We refer to our approach as Neural Interpolation (NeurInt) (Figure 1).

Ideally, we wish every point in the latent space to map to a unique, real image. However, it is unrealistic to expect a model to learn the entire data distribution over infinitely many real images given only a finite dataset [2]. Thus, given any image, we can only expect the model to learn to transform it to neighboring realistic images through suitable incremental changes. In view of this, instead of generating an image from random noise, our approach encourages the model to traverse through realistic images while maintaining smoothness and a net movement towards the target image.

---

[*]denotes equal contribution
[†]corresponding author

35th Conference on Neural Information Processing Systems (NeurIPS 2021), Sydney, Australia.

**Algorithm 1: NeurInt: Training**

**Input:** Dataset $P_{\text{data}}(\mathbf{x})$, Hyperparameter $\lambda$ and Integration time $T$

**1** Sample $\mathbf{x}_S$ and $\mathbf{x}_T$ from the dataset $P_{\text{data}}(\mathbf{x})$

**2** Set initial latent $\mathbf{z}_0$ of trajectory

$\qquad \mathbf{z}_0 := E(\mathbf{x}_S)$

**3** Sample Initial Velocity :

$\qquad \epsilon \sim \mathcal{N}(0, \mathbf{I})$

$\qquad \mathbf{v}_0 = \mu_v(\mathbf{z}_0, E(\mathbf{x}_T)) + \epsilon \odot \sigma_v(\mathbf{z}_0, E(\mathbf{x}_T))$

**4** Solve ODE System for for $z_t : [0, T] \longrightarrow \mathcal{X}$:

$\qquad \dfrac{d^2 \mathbf{z}_t}{dt^2} = f(\mathbf{z}_t, \dot{\mathbf{z}}_t) \quad \dot{\mathbf{z}}_0 = \mathbf{v}_0$

**5** Compute Reconstruction Loss

$\qquad \mathcal{L}_{\text{AE}} = \|\mathbf{x}_S - G(\mathbf{z}_0)\|_2^2 + \|\mathbf{x}_T - G(\mathbf{z}_T)\|_2^2$

**6** Sample $t_1, ..., t_N$ from Uniform $(0, T)$ without replacement

**7** Sample $\mathbf{x}_1, ..., \mathbf{x}_N$ from dataset

$\qquad \mathcal{L}_{\text{GAN}} = \sum_{i=1}^{N} \log(D(\mathbf{x}_i)) + \log(1 - D(G(\mathbf{z}_{t_i})))$

**8** Optimize the minimax game with (Stochastic) Gradient Descent Ascent

$\qquad \min\limits_{\mathcal{G}, \mathcal{E}, \mu_v, \sigma_v, f} \max\limits_{D} \; \mathcal{L}_{\text{GAN}} + \lambda \mathcal{L}_{\text{AE}}$

By *learning* to interpolate instead of using a deterministic interpolation technique, we allow the model to generate different categories of interpolations for different source and target images. This is achieved by learning a *distribution* over trajectories conditioned on the source and target images, parameterized by second-order Neural ODEs. Leveraging a *data-dependent* latent space distribution, parameterized through Neural ODEs, lends our approach the following major advantages: **1.** Direct utilization of real images while sampling latent vectors allows our approach to incorporate the benefits of non-parametric approaches like the ability to incorporate additional data into the generative model without retraining the parameters. **2.** Flexibility to learn different latent space distributions depending on the training data's complexity and size. **3.** The second-order formulation allows our model to map randomly sampled initial velocities from a simple Gaussian prior to a highly expressive class of smooth trajectories corresponding to different vector fields on the data distribution manifold. **4.** Continuous nature of the ODE allows us to sample an arbitrary number of points in each trajectory to obtain the desired level of smoothness in the interpolation trajectories where the smoothness is naturally enforced by the ODE formulation. **5.** Jointly enforcing smoothness and realism of interpolated images prevents the model from simply memorizing training data. **6.** Since our model's latent space distribution directly depends on the encoder, we do not require explicit matching of prior and posterior distributions, unlike other encoder-generator based approaches such as VAE [21] and ALI [10] **7.** The computational cost of sampling trajectories can be varied during training and test times by using different discretization schemes depending on the available computational resources.

## 2 Neural Interpolation (NeurInt)

Our approach, NeurInt, models a distribution over smooth continuous-time interpolation curves $\tilde{x}_t : [0, T] \longrightarrow \mathcal{X}$ (where $T \in \mathbb{R}^+$ and $\mathcal{X}$ represents the image manifold) which start from a given source image $\mathbf{x}_S$ and end at a target image $\mathbf{x}_T$. Similar to other latent variable-based generative models, we define the generated data distribution $p(\mathbf{x})$ as the distribution obtained by transforming a latent space distribution $p(\mathbf{z})$ through a generator $\mathcal{G}$. However, unlike generative models with fixed parametric priors, the latent space distribution $p(\mathbf{z})$ in our model is defined through a distribution over latent trajectories $\mathbf{z}_t : [0, T] \longrightarrow \mathcal{Z}$ conditioned over source and target images. Time evolution of these trajectories is governed by a Second-Order Neural ODE of the form $\dfrac{d^2 \mathbf{z}_t}{dt^2} = f(\mathbf{z}_t, \dot{\mathbf{z}}_t)$. In order to ensure that all image interpolation curves that are conditioned on $\mathbf{x}_S$ and $\mathbf{x}_T$ begin at the source, a *Position Encoder $E$* is used to project $\mathbf{x}_S$ to $\mathcal{Z}$, and the initial position $\mathbf{z}_0$ of the trajectory is set to $E(\mathbf{x}_S)$. The distribution over latent trajectories is defined by placing a data dependent prior on the initial velocity $\mathbf{v}_0$ or $\dot{\mathbf{z}}_0$. For our modeling purposes, we choose the prior $p(\mathbf{v}_0|\mathbf{x}_S, \mathbf{x}_T)$ to be a Diagonal Gaussian whose parameters are given by the *Velocity Encoder $\mathcal{V} = (\mu_v, \sigma_v)$*.

**Algorithm 2: NeurInt: Generation**

**Input:** Source Image $\mathbf{x}_S$, Target Image $\mathbf{x}_T$ and Integration time $T$
**Output:** Continuous-time Interpolation Curve $\tilde{\mathbf{x}}_t : [0, T] \longrightarrow \mathcal{X}$

**1** Set initial latent $\mathbf{z}_0$ of latent space trajectory
$$\mathbf{z}_0 := E(\mathbf{x}_S)$$
**2** Sample Initial Velocity :
$$\epsilon \sim \mathcal{N}(0, \mathbf{I})$$
$$\mathbf{v}_0 = \mu_v(\mathbf{z}_0, E(\mathbf{x}_T)) + \epsilon \odot \sigma_v(\mathbf{z}_0, E(\mathbf{x}_T))$$
**3** Solve ODE System for for $\mathbf{z}_t : [0, T] \longrightarrow \mathcal{X}$:
$$\frac{d^2\mathbf{z}_t}{dt^2} = f(\mathbf{z}_t, \dot{\mathbf{z}}_t) \quad \dot{\mathbf{z}}_0 = \mathbf{v}_0$$
**4** Generate Interpolation Curve $\tilde{\mathbf{x}}_t : [0, T] \longrightarrow \mathcal{X}$
$$\tilde{\mathbf{x}}_t = G(\mathbf{z}_t)$$

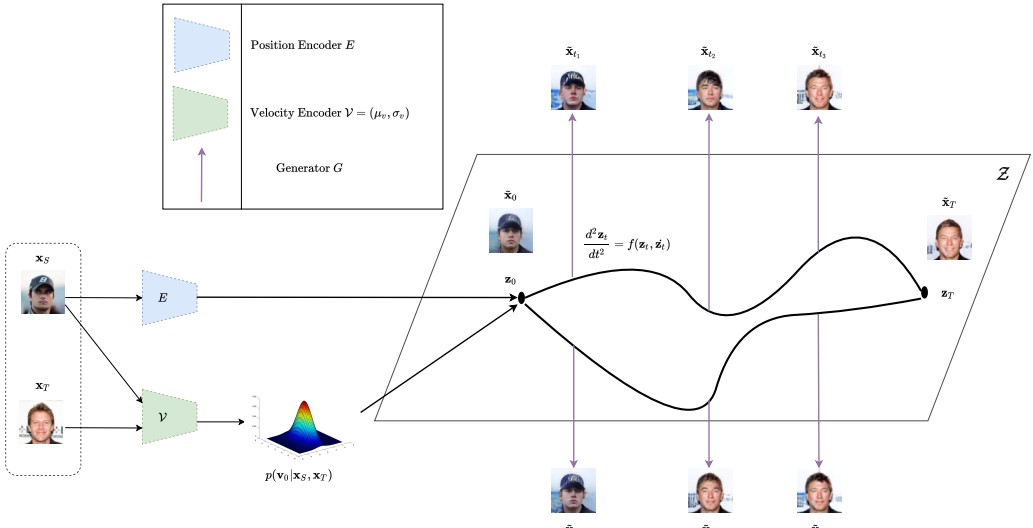

Figure 1: Generative Model of NeurInt. The deterministic position encoder $E$ projects the source image $\mathbf{x}_S$ to the latent space, the stochastic velocity encoder $\mathcal{V}$ inputs the source $\mathbf{x}_S$ and target $\mathbf{x}_T$, and outputs the parameters of a Diagonal Gaussian distribution over the initial velocity inducing a distribution over continuous-time latent interpolation trajectories $\mathbf{z}_t$ whose time evolution is governed by the Second-Order Neural ODE. The continuous-time interpolation curve $\tilde{\mathbf{x}}_t$ is generated by mapping $\mathbf{z}_t$ to the image manifold using the Generator $G$

Sampling a latent trajectory $\mathbf{z}_t$ hence consists of evaluating $\mathbf{z}_0$ and sampling an initial velocity $\mathbf{v}_0$ from the data dependent prior. This fixes the Initial Value Problem (IVP) for the trajectory, which can now be obtained by (numerical) integration of the ODE system. The true image space curve $\tilde{x}_t$ is then obtained by transforming $\mathbf{z}_t$ using the generator, and can be sampled at arbitrarily chosen time-points in the range $[0, T]$ to produce image samples.

The learning objective of the model ensures that a given image space curve $\tilde{\mathbf{x}}_t$ which is conditioned on a source target pair $(\mathbf{x}_S, \mathbf{x}_T)$ begins at the source and ends at the target. This is ensured by a pixel-MSE based reconstruction objective $\mathcal{L}_{\text{AE}}$ that matches $\tilde{\mathbf{x}}_0$ to $\mathbf{x}_S$ and $\tilde{\mathbf{x}}_T$ to $\mathbf{x}_T$. The realism and diversity of interpolation curves is ensured via Adversarial Learning. We use the Generative Adversarial Network to jointly train a critic $D : \mathcal{X} \longrightarrow [0, 1]$ which discriminates real samples drawn from the data distribution against evaluations of the interpolation trajectory at randomly sampled time-points . Learning is then formulated as a minimax game where the critic $D$ plays against the encoders $E$ and $\mathcal{V}$, generator $G$ and the Neural ODE $f$. The value function of the game is taken to be a weighted combination of the reconstruction and adversarial objectives. The entire training process is described in Algorithm 1.

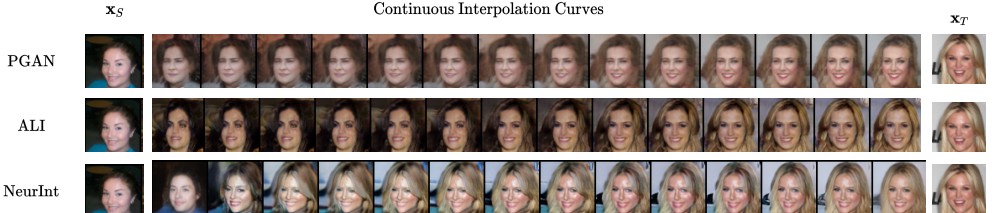

Figure 2: Example Interpolations on CelebA for Top : PGAN, Middle : ALI, Bottom : NeurInt. The first and last columns contain the source and target images respectively

# 3  Experiments

We benchmark NeurInt, which leverages learnable interpolation trajectories, against the interpolations generated by the Spherical (SLERP) and Linear Interpolations (LERP) on two base generative models, Progressive GAN (PGAN) [19] and Adversarially Learned Autoencoder (ALI) [10], on the Street View House Numbers (SVHN) [28] and CelebA [24] datasets. To maintain uniformity, the architecture of the Generator and Discriminator for all three models, and that of the Encoder for ALI and NeurInt resemble a standard Progressive GAN. The Neural ODE component of NeurInt uses a Runge Kutta (RK4) integrator with 32 integration timesteps and a total integration time of 1 second ($T = 1$). Further details regarding our architectural choices and hyperparameter selection are described in the Appendix. To evaluate our approach's interpolation capabilities compared to the baselines, we project the source and target images in the latent space of the respective models, and compare the continuous-time learned interpolations of NeurInt against the interpolations generated by SLERP and LERP for each of the baseline generative models. For NeurInt and ALI, projecting the image to an encoding space is trivial since both the models jointly learn an encoder from the image space to the latent space. However, since PGAN lacks any such means of projection, we train an encoder $E_P$ that learns to project images onto the trained Progressive GAN latent space. The architecture of $E_P$ is the same as that of the encoder $E$ used by NeurInt and it is trained by minimising a pixel-wise MSE loss and the encoder is progressively grown to maintain consistency.

**Sample Quality, Diversity & Incorporating Unseen Data** We quantitatively assess the generation and interpolation quality for NeurInt and our baselines using Frechet Inception Distance (FID) [15], a standard evaluation metric for GANs, and present the results in Table 5. We randomly select 5000 pairs of source-target images from a support distribution. For NeurInt we generate interpolation trajectories for each pair and randomly sample two intermediate interpolants from each trajectory. For the PGAN and ALI baselines, we project each source-target pair into the latent space, and then generate trajectories using Spherical and Linear Interpolation. The FIDs so obtained are listed under LERP and SLERP in Table 5. To decouple the evaluation of interpolation quality from that of sample generation, we also evaluate the FID of the baselines using samples drawn from their true generative model (listed as PGAN-PRIOR and ALI-PRIOR in Table 5), by sampling a latent code from their respective priors.

**Adapting to Unseen Supports** Since our approach models distribution over trajectories conditioned on source and target images, the generated data distribution can be flexibly varied by modifying the distribution of source and target images. This allows the trained model to improve the generated data's diversity without retraining the parameters by incorporating additional data into the set of source and target images. This is unlike the models based on fixed parametric priors, which require retraining on new data to modify the generated data distribution. We repeat the evaluation by varying the support across 3 different distributions, namely the Train Set, the Test Set, as well as the Train and Test set combined. As demonstrated through the results in Table 1, utilizing additional data from the test set leads to improvement in FID scores. Our model's ability to interpolate on test data also demonstrates its ability to model the entire image data manifold rather than overfitting on training images.

**Varying ODE Solver Configuration** The Second-Order Neural ODE formulation of NeurInt allows it to interpolate in continuous time. This imparts our approach the flexibility of varying the the level of discretization (number of time-steps) as well as the ODE solver at test time. We validate this by generating trajectories while reducing the number of integrator steps from 32 to 20 in steps of 4 for both RK4 and Euler algorithms. For a fair comparison, we correspondingly vary the time-resolution of the LERP and SLERP interpolators of our baselines and benchmark the models using the FID metric. The results are presented in Table 3. We observe that even at very low integration time-steps,

Table 1: FID scores ($\downarrow$) across various supports & sampling methods.

| SUPPORT | DATASET | METHOD | PGAN | ALI | NEURINT |
|---|---|---|---|---|---|
| TRAIN SET | SVHN | PRIOR | 17.05 | 46.45 | |
| | | LERP | 25.23 | 36.59 | **6.45** |
| | | SLERP | 25.43 | 36.52 | |
| | CELEBA | PRIOR | 11.57 | 19.24 | |
| | | LERP | 36.28 | 24.21 | **10.23** |
| | | SLERP | 36.45 | 24.25 | |
| TEST SET | SVHN | PRIOR | 17.05 | 46.45 | |
| | | LERP | 29.53 | 37.61 | **6.82** |
| | | SLERP | 30.06 | 37.17 | |
| | CELEBA | PRIOR | 11.57 | 19.24 | |
| | | LERP | 35.94 | 23.78 | **10.37** |
| | | SLERP | 36.63 | 23.38 | |
| TRAIN & TEST SET | SVHN | PRIOR | 17.05 | 46.45 | |
| | | LERP | 24.74 | 37.57 | **6.24** |
| | | SLERP | 25.31 | 37.04 | |
| | CELEBA | PRIOR | 11.57 | 19.24 | |
| | | LERP | 36.27 | 23.77 | **10.17** |
| | | SLERP | 36.43 | 23.95 | |

Table 2: FID scores ($\downarrow$) on CelebA dataset across various models & sampling methods.

| METHOD | PGAN | ALI | NEURINT | NEURINT-PT | LINT | SINT | FOINT I | FOINT II |
|---|---|---|---|---|---|---|---|---|
| PRIOR | 11.57 | 19.24 | | | | | | |
| LERP | 36.27 | 23.77 | **10.17** | 16.94 | 16.5 | 13.2 | 24.5 | 13.5 |
| SLERP | 36.43 | 23.95 | | | | | | |

the RK4 variant of NeurInt consistently outperforms the baselines. The same does not hold true for the Euler integrator for very low timesteps, which could be attributed to the inherent coarseness and piecewise linear nature of the Euler Integrator. The discretization invariance of NeurInt has immense practical utility, as it allows us to train the model at a very fine time resolution on a powerful hardware configuration, while deploying it at test time on less powerful hardware by reducing the accuracy and integrator timesteps of the solver. We also report the time taken for generating the latent interpolants in Table 3 for NeurInt as well as LERP and SLERP in PGAN. We note that RK4, despite using four intermediate increments per solver step, is consistently faster than SLERP.

Table 3: FID scores ($\downarrow$) on CelebA and time on varying the solver and timesteps.

| Steps | FID | | | | | | Time(s) for 64000 images | | | |
|---|---|---|---|---|---|---|---|---|---|---|
| | NeurInt | | PGAN | | ALI | | NeurInt | | PGAN | |
| | RK4 | Euler | LERP | SLERP | LERP | SLERP | RK4 | Euler | LERP | SLERP |
| 20 | 10.63 | 12.23 | 36.00 | 36.73 | 24.18 | 24.26 | 41.55 | 15.47 | 12.96 | 60.19 |
| 24 | 10.47 | 11.90 | 36.49 | 36.85 | 24.24 | 24.26 | 49.85 | 23.84 | 20.81 | 79.69 |
| 28 | 10.43 | 11.23 | 36.29 | 37.25 | 24.04 | 24.12 | 58.54 | 33.94 | 40.74 | 113.87 |
| 32 | 10.37 | 10.94 | 35.94 | 36.63 | 23.78 | 23.38 | 62.23 | 40.75 | 54.11 | 119.43 |

**Ablation Study** To evaluate the benefits of jointly incorporating smooth interpolation on the latent space through the use of second-order Neural ODEs and non-parametric data-dependent prior on the latent space obtained by conditioning the generated images on the randomly sampled source and target images, we perform an ablation experiment ( NeurInt-PT) where we train the generative model to map images from a fixed latent space prior using the original Generative Adversarial Networks framework and subsequently utilize a Neural ODE network to learn realistic interpolation trajectories on the fixed latent space. In Table 5 we report the FID of NeurInt-PT.

**Choice of using a $2^{\text{nd}}$ Order ODE** Free initial velocity parameter in $2^{\text{nd}}$ Order ODE allows us to parameterize a trajectory distribution for every source target pair. Such a parameter is absent in fixed interpolation schemes and $1^{\text{st}}$ Order ODEs of the form $\frac{d\mathbf{z}_t}{dt} = f(\mathbf{z}_t)$ or $\frac{d\mathbf{z}_t}{dt} = f(\mathbf{z}_t, \mathbf{z}_C)$ where $\mathbf{z}_C \sim \mathcal{N}(\mu_v(\mathbf{z}_0, E(\mathbf{x}_T)), \sigma_v(\mathbf{z}_0, E(\mathbf{x}_T))^2\mathbf{I})$. Hence, using LERP, SLERP or a $1^{\text{st}}$ Order ODE in Algo.1 Step 4 would prevent us from learning a trajectory distribution for a source-target pair, since such approaches uniquely fix a trajectory given two endpoints and we quantitatively justify our choice by training the corresponding models called LINT, SLINT, FOINT I and FOINT II. As shown by FID scores in Table 5, NeurInt outperforms these models.

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

## A   Related Work

The progress in the design of latent variable-based deep generative models such as GANs [12], VAEs [21], and normalizing flows [31] in recent years has led to numerous applications. Generative

Adversarial Networks (GANs), in particular, have been extensively utilized for image generation [19, 4, 9, 18], video generation [36, 6, 8], image translation [17, 41], as well as various other tasks requiring the generation of high-dimensional data. This has also led to vast literature on improving inference [10, 7, 25, 37, 26], and training stability [27, 1] of GANs. However, work on improving the diversity, smoothness, and realism of interpolations has been limited. One of the major reasons for poor interpolation quality is the mismatch between latent vectors' distributions corresponding to interpolations and the prior distribution used during training. Some recent works have attempted to tackle this mismatch by modifying the prior [38, 22] or using non-parametric priors [33]. However, unlike our approach which ensures matching image distributions corresponding to entire trajectories with real data distribution, these works only focus on matching the latent space distribution for midpoints of sampled noise vectors. A recent work [5] exploited Neural Ordinary Differential Equations' inherently sequential nature to model continuous time dynamics of time-series data. However, such an approach cannot be directly applied to generating interpolations in the training data. Therefore, we leverage Neural ODEs [5] for traversing the latent space while ensuring realness through the use of a discriminator. Our work is also related to but different from Exemplar models [29] and Kernel Density Estimation [30]. While Exemplar based generative models directly utilize the dataset for modeling the latent space distribution, each generated image in such models is obtained by modifying only one randomly sampled image. Our approach instead can find points on image space between any arbitrarily pair of images, enabling it to capture the full diversity of image data manifolds for both generation as well as interpolation. Moreover, our GAN based formulation allows directly matching the image distribution of trajectories of interpolants with the real data distribution. This obviates the need of utilizing a nearest-neigbour based approximation of a closed form non-parametric distribution in the latent space. By directly enforcing smoothness and realism of interpolated images, our approach also prevents memorization of training data without utilizing regularization [29] or pseudo-inputs [35].

The set of source and target images in our model can also be interpreted as a form of external memory, which has been shown to improve generation quality in several works [3, 14, 16, 34, 20, 23].

The use of Neural ODEs in our approach is inspired by their recent application to a variety of domains. On account of their continuous-time formalism, Neural ODEs, unlike traditional discrete-time sequence models, are capable of handling non-uniformly sampled temporal data with no additional overhead. As a result, they are inherently suited to applications, such as time-series forecasting [5, 32] and deep generative models of continuous-time data [5, 39], moreover, due to their smoothness and invertibility properties [40], Neural ODEs also find application in density estimation and variational inference as Continuous Normalizing Flows. [5, 13]

## B   Background

Since our approach is based on neural ODEs, in this section, we briefly review neural ODEs and second-order neural ODEs, and approaches to solve these.

**Neural ODEs** Various deep learning architectures, such as Residual Networks, RNNs and Normalizing Flows can be be formulated as a discrete sequence of additive transformations on a state variable $\mathbf{z}_t$

$$\mathbf{z}_{t+1} = \mathbf{z}_t + f_\theta(\mathbf{z}_t)$$

where the transition function $f_\theta$ is modelled by a neural network. The above operation can be identified as a unit time-step Euler discretization of a continuous-time system. Hence, taking the continuous limit of the additive transition, we obtain a First-Order ODE system for $\mathbf{z}_t$

$$\frac{d\mathbf{z}_t}{dt} = f_\theta(\mathbf{z}_t)$$

The neural network $f_\theta$, which was previously the discrete-time transition function, now becomes the vector field for the First-Order ODE governing the time evolution of the state $\mathbf{z}_t$. This framework is known as Neural Ordinary Differential Equations [5]. The value of the state $\mathbf{z}_t$ at any given time, as a function of the input or initial state $\mathbf{z}_0$, is obtained by solving the Initial Value Problem (IVP)

$$\mathbf{z}_t = \mathbf{z}_0 + \int_0^t f_\theta(\mathbf{z}_\tau)d\tau$$

Table 4: Misclassification rate ($\downarrow$) on the test set of SVHN demonstrating the usefulness of learned representations.

| Model | Misc rate (%) |
|---|---|
| PGAN | $0.3254 \pm 0.0024$ |
| ALI | $0.2848 \pm 0.0840$ |
| NeurInt | $\mathbf{0.2647 \pm 0.0018}$ |

While exact solution is infeasible in most cases, the IVP can be approximately solved with high accuracy using Numerical ODE solvers such as Runge Kutta (RK4) and Dormand Price (DOPRI5). Gradients can either be obtained by ordinary backpropagation or by using the Adjoint State Method, [5] which allows gradient computation without backpropagating through ODE solver operations.

**Second-Order Neural ODEs** Despite the impressive continuous-time modeling capabilities of (First-Order) Neural ODEs, there exist various classes of phenomena (e.g. Harmonic and Van der Pol oscillators) whose latent dynamics cannot be modeled by First-Order ODE systems. This motivates the use of Second-Order Neural ODEs [39], a framework where the time evolution of the state variable $\mathbf{z}_t$ is governed by

$$\frac{d^2\mathbf{z}_t}{dt^2} = f_\theta(\mathbf{z}_t, \dot{\mathbf{z}}_t)$$

Analogous to First-Order Neural ODEs, the vector field $f_\theta(\mathbf{z}_t, \dot{\mathbf{z}}_t)$ is modelled by a Neural Network. However, a key difference lies in the fact that the vector field is a function of the state $\mathbf{z}_t$ as well as the state differential $\dot{\mathbf{z}}_t = \frac{d\mathbf{z}_t}{dt}$. This feature allows Second-Order Neural ODEs to model much more complex dynamical systems that cannot be modeled by First-Order Neural ODEs. Moreover, Second-Order Neural ODEs have much better smoothness properties as they ensure the continuity of the second derivative of state $\frac{d^2\mathbf{z}_t}{dt^2}$

To facilitate numerical integration, the Second-Order Neural ODE is reduced to an equivalent Coupled First-Order ODE system by introducing an auxiliary state variable $\mathbf{v}_t$ (often named velocity) as follows.

$$\frac{d\mathbf{z}_t}{dt} = \mathbf{v}_t \quad \frac{d\mathbf{v}_t}{dt} = f_\theta(\mathbf{z}_t, \mathbf{v}_t)$$

This ODE system can be interpreted as a First-Order Neural ODE for the augmented state $[\mathbf{z}_t, \mathbf{v}_t]^T$. Consequently, it can be transformed into an Initial Value Problem (IVP)

$$\begin{bmatrix} \mathbf{z}_t \\ \mathbf{v}_t \end{bmatrix} = \begin{bmatrix} \mathbf{z}_0 \\ \mathbf{v}_0 \end{bmatrix} + \int_0^t \begin{bmatrix} \mathbf{v}_\tau \\ f_\theta(\mathbf{z}_\tau, \mathbf{v}_\tau) \end{bmatrix} d\tau$$

which, as described in Section 3.1, can be solved by Numerical ODE solvers. As discussed earlier, gradient computation can be performed either by backpropagating through the ODE solver's operations or by using the Adjoint State Method [5].

## C    Additional Experiments

**Improved Representation Learning** We evaluate the representation learning capabilities of NeurInt and our baselines by training a linear SVM model on the feature vectors obtained by concatenating the output layer and the last hidden layer of the encoder for 60,000 class balanced labeled images from the training set of the SVHN dataset. We hold out 10,000 labeled images from the training set as a validation set to tune hyper-parameters of the SVM model. We report the average test misclassification error for 10 different SVM models trained on different random 60,000 example training sets. Our results are reported in Table 4.

**Learning a Distribution of Trajectories** To assess the diversity of the interpolation trajectories resulting from the distribution modeled by NeurInt, we generate different trajectories by fixing a source-target pair, drawing multiple samples of $\mathbf{v}_0$ conditioned on this fixed source-target pair and generating the corresponding interpolation trajectories. As observed in Rows $A_1$ and $A_2$ of Figure 3, the sampled trajectories show noticeable variation in the intermediate interpolants but successfully converge to the same target, as desired. To further emphasise on this variation, we repeat the process by using the same source as Rows $A_1$ and $A_2$ but using a different target image as shown in Rows $A_3$

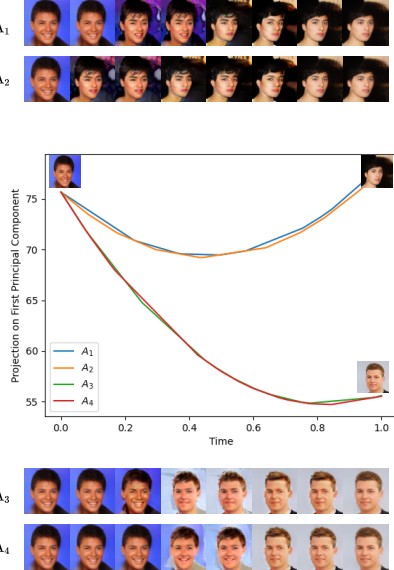

Figure 3: NeurInt learning a distribution of trajectories for interpolation. The top two rows represent samples on the trajectories $A_1$ and $A_2$, while the bottom two rows represent samples on trajectories $A_3$ and $A_4$.

and $A_4$. To visualize these variations in the encoding space, we plot the first Principal Component of each row of Figure 3 over time. It is observed in the plot (Figure 3) that the PCA [11] curves of trajectories $A_1$ and $A_2$, starting from the same source, deviate from one another in the middle, thereby reflecting the variety of intermediate interpolants, and towards the end, converge very close to each other, in the neighborhood of the target. The same phenomenon is observed for $A_3$ and $A_4$, whose PCA curves deviate significantly from $A_1$ and $A_2$, on account of having a different target. Furthermore, the PCA curves' curvature and smoothness confirm that NeurInt truly captures a distribution of smooth and non-linear interpolation trajectories.

# D   Architecture and Setup

For our baselines and the proposed model, we borrow the architecture for generator and discriminator from PGAN [19]. For ALI, following [10], we use two networks $D_{\mathcal{X}}$ and $D_{\mathcal{Z}}$ to extract features from a given image $X$ and latent vector $Z$ respectively which are subsequently concatenated and passed through a joint network $D_{\mathcal{X},\mathcal{Z}}$ to obtain $D(X, Z)$. To ensure fairness, all models were trained progressively. The encoder $E$'s architecture uses the same layers as the first 12 layers of the discriminator architecture's $D_{\mathcal{X}}$ component.

For NeurInt, the position Encoder $E$ and velocity encoder $\mathcal{V} = (\mu_{\mathbf{v}}, \sigma_{\mathbf{v}})$ are both one hidden layer MLPs with a LeakyReLU nonlinearity, and the the vector field $f$ of the Second Neural ODE is a 2 layer MLP with a tanh nonlinearity. The relative weighting hyperparameter $\lambda$ in the loss for NeurInt was decayed linearly from 1000 to 100 per half cycle of each progressive step and then kept stable at 100 for the next half-cycle of the step.

To maintain consistency, we also trained the encoder $E_P$ for PGAN by growing it progressively. We obtained an RMSE error of 0.0522 on the held-out test set upon training. Figure 4 shows some reconstructions on held-out test data which qualitatively indicate the convergence of the encoder.

We use the same optimizer for PGAN and Neurint as proposed in [19] and for ALI as in [10].

For all our experiments, we use 2 Nvidia GeForce GTX 1080 Ti GPUs.

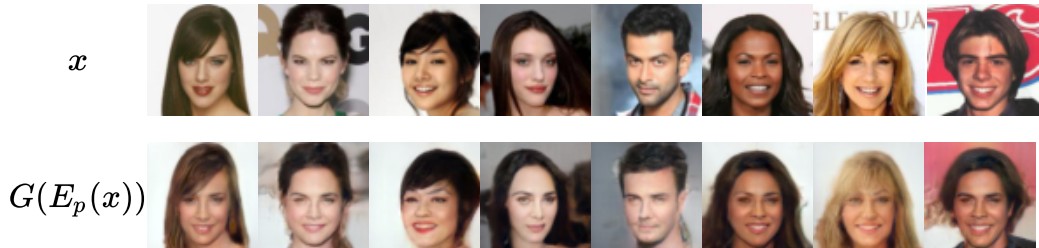

$x$

$G(E_p(x))$

Figure 4: Sample Reconstructions of Encoder $E_p$

Table 5: FID Score Statistics for NeurInt, PGAN and ALI on CelebA and SVHN. Lower FID is better. For all sampling methods other than PRIOR, the training set is used as the support

| DATASET | METHOD | PGAN | ALI | NEURINT |
|---------|--------|------|-----|---------|
| SVHN | PRIOR | 17.05±0.082 | 46.45±0.075 | |
| | LERP | 25.23±0.072 | 36.59±0.073 | **6.45±0.069** |
| | SLERP | 25.43±0.081 | 36.52±0.066 | |
| CELEBA | PRIOR | 11.57±0.088 | 19.25±0.0.085 | |
| | LERP | 36.29±0.078 | 24.20±0.090 | **10.22±0.082** |
| | SLERP | 36.44±0.081 | 24.25±0.091 | |

## E  FID Statistics

Using the training set as the support, we compute the FID scores of NeurInt and our baselines (using the same procedure as described in the main paper), over 100 randomized runs, and report the mean and standard deviation of the FID scores obtained in Table 5.

## F  Additional Samples

**Interpolation Samples :**  Figures 5 and 6 qualitatively demonstrate NeurInt's ability to generate significantly more realistic and smoother interpolation trajectories over the baselines.

**Uncurated Samples :**  Figure 7 shows uncurated samples from NeurInt and baselines. The quality of the generated samples of NeurInt against the baselines backs up the superior FID achieved by NeurInt. This demonstrates NeurInt to be not only a good interpolation methodology but also a good generative model.

**Distribution of trajectories :**  Figure 8 demonstrates qualitatively NeurInt's ability to draw interpolations from a distribution of interpolation trajectories. Particularly note the transition between a female source and male target with different facial attributes.

**Ablation Study :** Figure 9 demonstrates the benefits of our joint training mechanism, with NeurInt interpolation trajectory being of significantly better quality than NeurInt-PT.

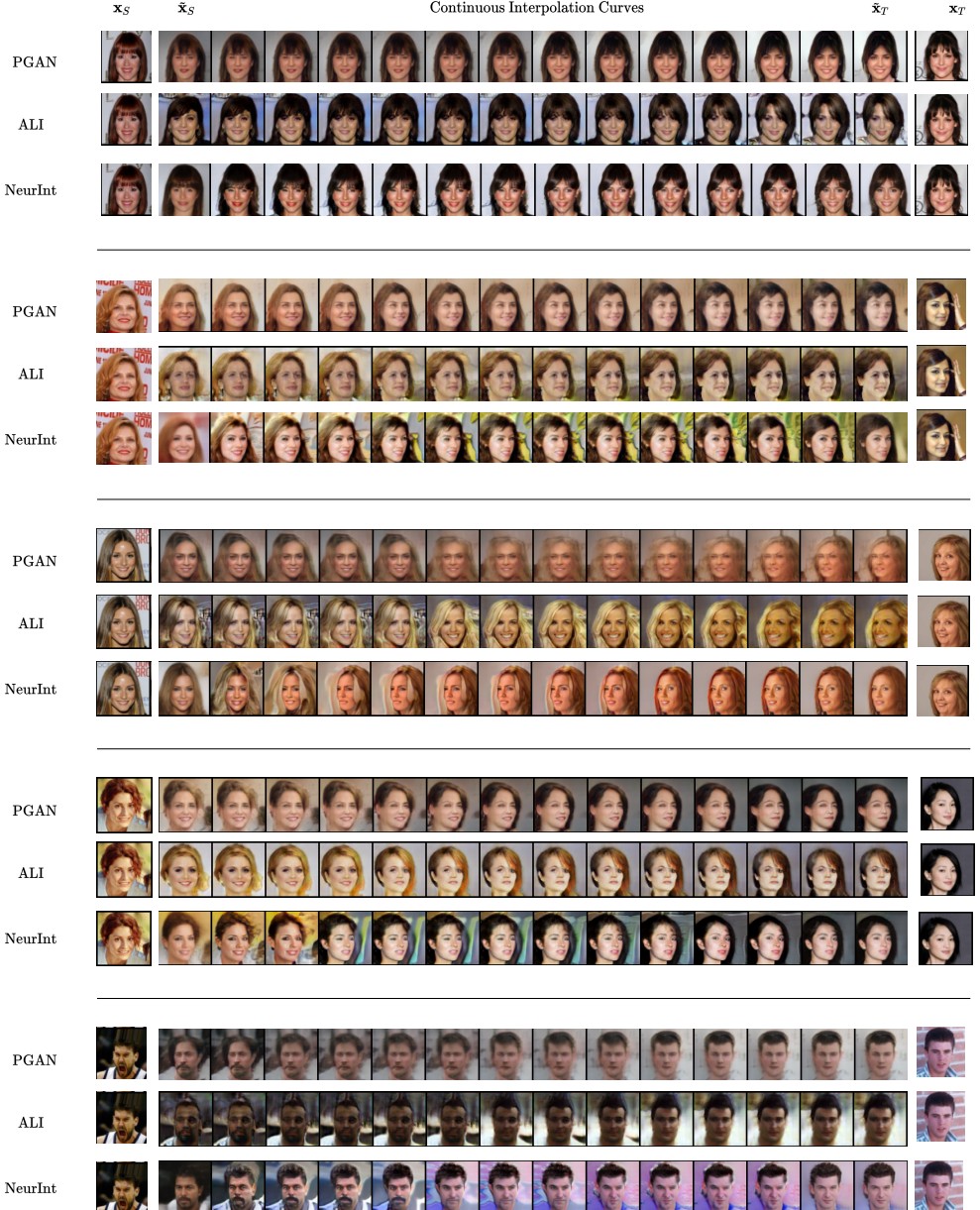

Figure 5: Comparison of Interpolation quality between PGAN, ALI and NeurInt on the CelebA dataset. $\mathbf{x}_S$ (leftmost) and $\mathbf{x}_T$ (rightmost) denote the true source-target pair from the training set on which the trajectory was conditioned. The interpolation trajectories (shown in the middle) begin at $\tilde{\mathbf{x}}_S$ (reconstruction of $\mathbf{x}_S$) and end at $\tilde{\mathbf{x}}_T$ (reconstruction of $\mathbf{x}_T$)

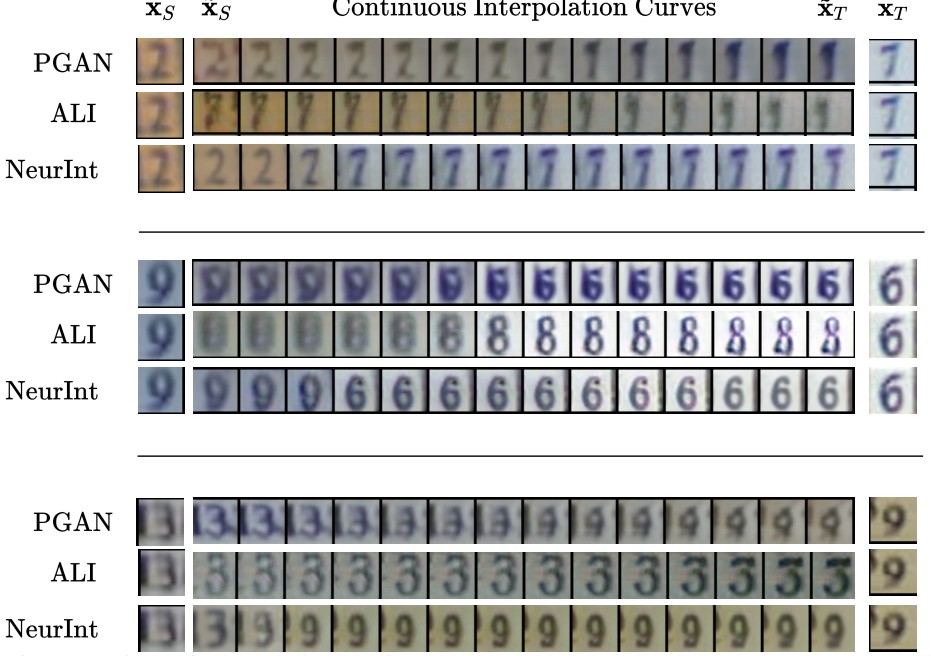

Figure 6: Comparison of Interpolation quality between PGAN, ALI and NeurInt on the SVHN dataset. $\mathbf{x}_S$ (leftmost) and $\mathbf{x}_T$ (rightmost) denote the true source-target pair from the training set on which the trajectory was conditioned. The interpolation trajectories (shown in the middle) begin at $\tilde{\mathbf{x}}_S$ (reconstruction of $\mathbf{x}_S$) and end at $\tilde{\mathbf{x}}_T$ (reconstruction of $\mathbf{x}_T$)

Progressive GAN                    ALI                        NeurInt

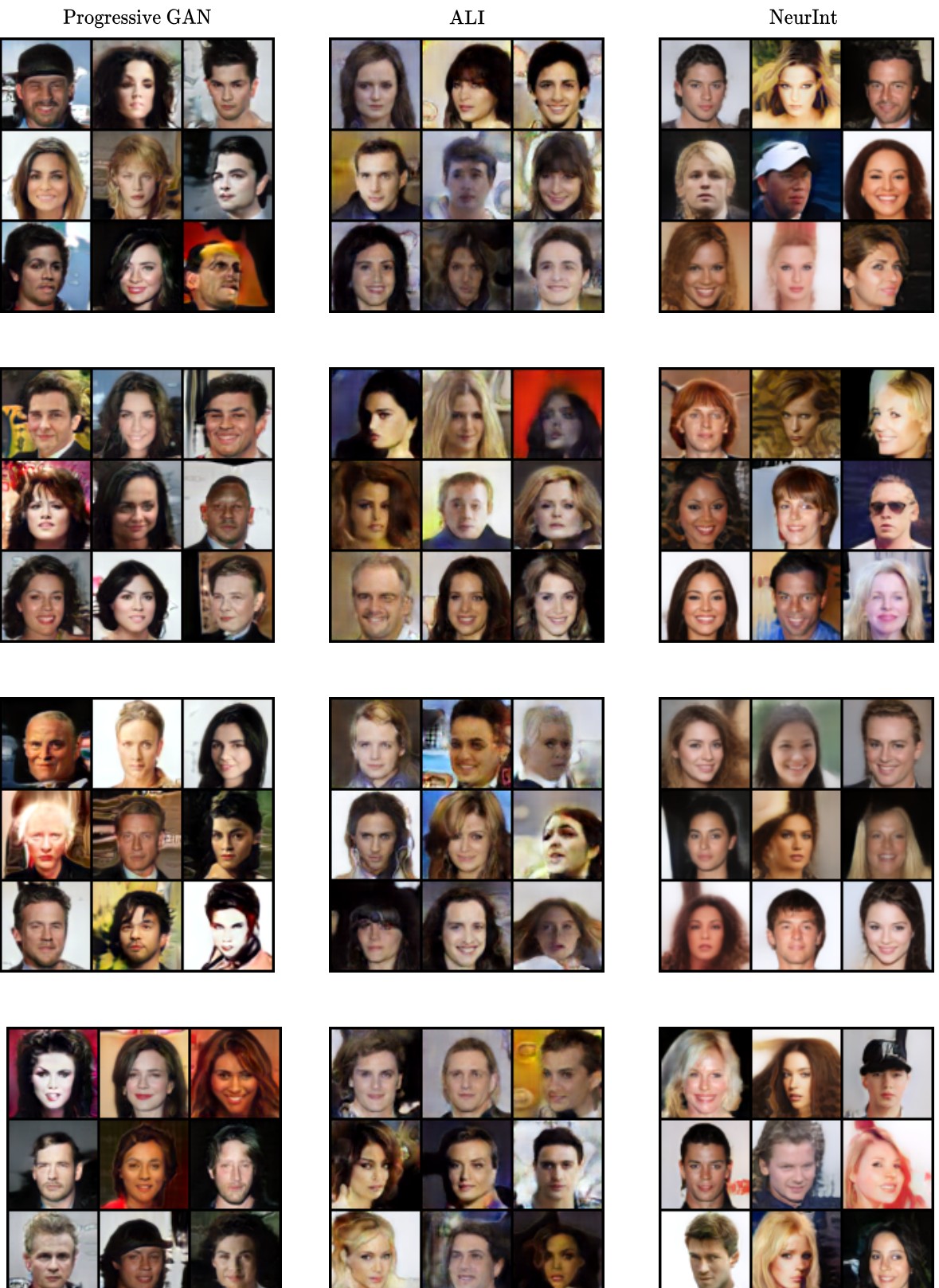

Figure 7: Uncurated samples from Progressive GAN (left), ALI (middle), and NeurInt (right) trained on the CelebA dataset. Samples from Progressive GAN and ALI are drawn from their true prior distribution, whereas samples from NeurInt are drawn by first generating continuous-time trajectories and then evaluating them at random intermediate points

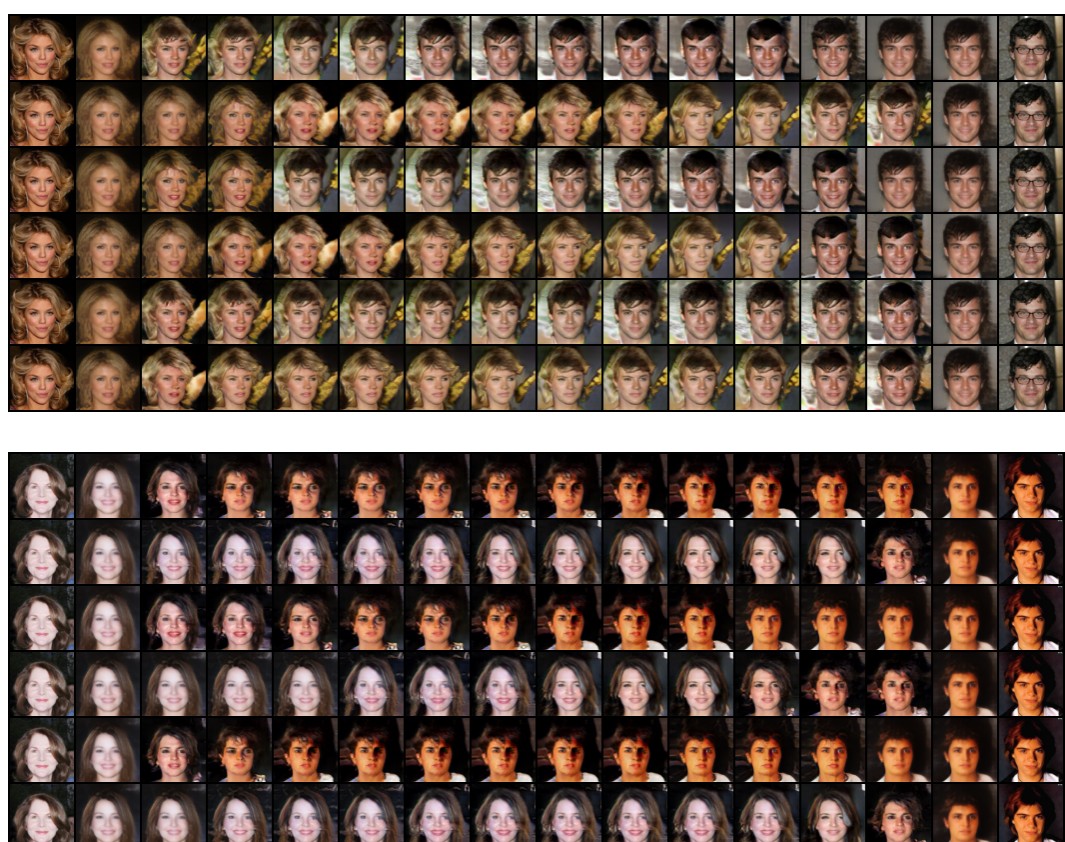

Figure 8: NeurInt's samples upon choosing different random initial velocities demonstrating the model's ability to learn a distribution of trajectories. Each row is an interpolation between the real images on the first and last columns.

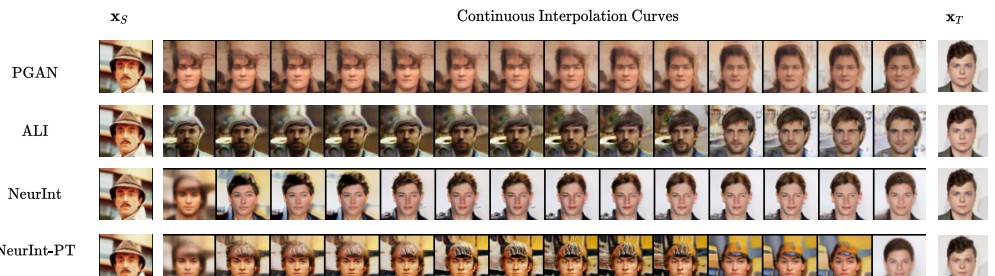

Figure 9: Example Interpolations for PGAN, ALI, NeurInt and NeurInt-PT. The first and last columns contain the source and target images respectively

