# OpenReview forum: "NeurInt: Learning to Interpolate through Neural ODEs"
_NeurIPS.cc/2021/Workshop/DLDE — DLDE Workshop -- NeurIPS 2021 Spotlight_

### Official Review · Reviewer_cET4 · 2021-09-29
**Commendable attempt at designing a new generative model which learns a trajectory distribution through SONODEs, with reliance on deterministic interpolation method**

**Confidence:** 2

**Review:**

In this work, the authors present a new generative model, NeurInt, which uses a combination of reconstruction and adversarial losses to learn a distribution of trajectories between 2 images using Latent SONODEs. The use of DE to learn a set of trajectories is a complicated avenue. On one hand, parameter setting and gaining insights about a random test sample becomes easy. On the other hand, a lot of complications (like biases and SONODE parameter selection ) arise when using the same. This paper focuses on leveraging a data-dependent latent space distribution, which offer up tempting advantages. On the whole, NeurInt is shown to be well capable to learning distribution and explores interesting avenues. Some comments:

1) Occasional grammatical errors can be overlooked, but glaring errors like the title "Introdcution" really draw attention to themselves

2) In the results discussion, I noticed that the datasets used are quite large and the FID metric was used to measure NeurInt's goodness-of-fit. FID is known to display high bias, when conducted on smaller datasets. Hence, although the choice of datasets is well and good, addition of other metrics like IS and FVD could enhance this work.

3) What would happen if augmented neural ODEs were used instead of SONODEs? Would they decrease the complexity or increase hyper parameter selection headache?



**Score:**

3: Good paper

---

### Official Review · Reviewer_K2oK · 2021-09-29

**Confidence:** 4

**Review:**

The authors propose a generative model over continuous, interpolating paths between a source and a target image. The approach takes the form of a deterministic autoencoder tasked to produce realistic interpolating images in the decoded trajectory $z_t$ via adversarial training.

**Novelty and Significance:**

The work represents a creative application of latent neural ODE variants such as ODE$^2$VAE [1]. The differences are substantial, including specifics of the model itself and target application. There are several downstream problems that can be solved with a flexible generative model directly over continuous interpolating paths between data points, such as generating realistic videos.

**Questions:**

Some general questions that the authors might want to consider to extend their work or tease out additional properties of their approach:

* Have the authors investigated what would happen if a realistic-looking interpolation between source and target does not exist? Depending on the $\lambda$ the model might prefer to stick to the constraint at $z_t$ or instead choose to generate realistic images in an interpolating path that does not end at the target image.
* The second-order ODE choice is slightly unusual for this task. I was wondering if the authors considered simpler ways to condition with target image information, such as data-controlled neural ODEs [2].
* An integral adversarial loss [2] via a generalized adjoint method might further improve the quality of images across the interpolation.
* How performant is this model as a generic generative model over images? How would you go about sampling without a target image -- for example feeding the same image as both source and target?

[1] ODE2VAE: Deep generative second order ODEs with Bayesian neural networks, Yildiz et al NeurIPS2019
[2] Dissecting Neural ODEs, Massaroli et al NeurIPS20


**Score:**

3: Good paper

---

### Official Review · Reviewer_wBJ8 · 2021-10-11
**Review: Nice work, some comments**

**Confidence:** 3

**Review:**

**Summary:**

The paper focuses on learning the interpolation between two images that stays close to the image manifold. Smooth dynamics are imposed on a lower dimensional encoding space using a second-order neural ordinary differential equation to generate the trajectory conditioned on the end-point images (source and target). Loss is constructed of a weighted sum of a reconstruction loss between the pair of the sampled end-point images and the discriminator loss of the sample images from the trajectory. The experiments are performed on the CelebA dataset and the street view house numbers dataset showing promising results.

**Main review:**

I am not well-aware of the related literature, so I will trust other reviewers on the novelty of the method. However, I have a couple of comments regarding the method and the results.

1. What is the main advantage of using a second-order neural ODE? Will learning an SDE provide a more robust framework to learn the latent trajectory?
2. The noise is introduced only on the initial state. It might result in a high variability of the initial state but less so at the end of the trajectory. Was any such issue faced by the authors? In Figure 2, your interpolation does a single big jump initially which lands on the target straight away.
3. A brief description or a reference for LINT, SINT, FOINTI, and FOINTII would help the unfamiliar reader.
4. L342 states that the position encoder and the velocity encoder are one hidden layer MLPs? So, if I understood it correctly, going from the observed high-dimensional image space to the encoded space is through one hidden layer MLP? Is the encoder able to extract all the useful features? How about alternative architectures like convolution?

**Minor comments:**

1. In Algorithm 1, step 8, the symbols mismatch over which the model is minimized.
2. Some other minor typos like on L15, L111, the caption of Figure 3, etc.
3. Just a suggestion, separate references for the main paper and appendix.
4. Bibliography has et al. but should have all the author names; something worth checking?
5. Worth mentioning in Table2 the dataset for which the results are presented.
6. The main paper talks about experiments on street view house numbers dataset, but the main paper does not show any result for it?

**Score:**

3: Good paper

---

### Decision · Program_Chairs · 2021-10-17

**Decision:**

Accept (Spotlight)

**Comment:**

This paper consider a second-order neural ODE to generate a latent-space trajectory as an alternative to linear or spherical interpolation. It has received positive reviews and is a good contribution to the workshop as an example of how differential equations can be useful for constructing novel operations within latent-variable models. The reviewers have posed some interesting questions about this work that would make a good contribution to a spot-light talk.

One improvement that would add to this paper is to take the work of [1] into consideration in addition to linear or spherical interpolation.

[1] Optimal transport maps for distribution preserving operations on latent spaces of Generative Models. Eirikur Agustsson et al. ICLR 2018.